# Effectiveness of Point of Entry Health Screening Measures among Travelers in the Detection and Containment of the International Spread of COVID-19: A Review of the Evidence

**DOI:** 10.3390/ijerph21040410

**Published:** 2024-03-28

**Authors:** Remidius Kamuhabwa Kakulu, Esther Gwae Kimaro, Emmanuel Abraham Mpolya

**Affiliations:** 1Department of Health and Biomedical Sciences, School of Life Science and Bioengineering, Nelson Mandela African Institution of Science and Technology (NM-AIST), Arusha P.O. Box 447, Tanzania; esther.kimaro@nm-aist.ac.tz (E.G.K.); emmanuel.mpolya@nm-aist.ac.tz (E.A.M.); 2Department of Preventive Services, Ministry of Health, Dodoma P.O. Box 743, Tanzania; 3Institute of Biodiversity, Animal Health & Comparative Medicine, University of Glasgow, Glasgow G12 8QQ, UK; 4Center for Global Health (CGH), Perelman School of Medicine University of Pennsylvania, 240 John Morgan Bldg., 3620 Hamilton Walk, Philadelphia, PA 19104, USA; 5Institute for Health Metrics and Evaluation (IHME), Population Health Building/Hans Rosling Center, 3980 15th Ave. NE, Seattle, WA 98195, USA

**Keywords:** systematic review, points of entry, airport, port, border, screening, COVID-19

## Abstract

COVID-19 remains a communicable disease with the capacity to cause substantial damage to health and health systems. Enhanced health screening at points of entry (POEs) is a public health measure implemented to support early detection, prevention and response to communicable diseases, such as COVID-19. The purpose of this study was to review the available evidence on the effectiveness of POE health screening in the detection and containment of the COVID-19 pandemic. This study was registered under PROSPERO and followed PRISMA guidelines in which the literature between 2019 and 2022 was retrieved from Scopus, PubMed, Web of Science, Global Health, CINAHL, Embase, Google Scholar and international organizations. A total of 33,744 articles were screened for eligibility, from which 43 met the inclusion criteria. The modeling studies predicted POE screening able to detect COVID-19 in a range of 8.8% to 99.6%, while observational studies indicated a detection rate of 2% to 77.9%, including variants of concern depending on the screening method employed. The literature also indicated these measures can delay onset of the epidemic by 7 to 32 days. Based on our review findings, if POE screening measures are implemented in combination with other public health interventions such as rapid tests, they may help detect and reduce the spread of COVID-19.

## 1. Introduction

Severe acute respiratory syndrome coronavirus 2 (SARS-CoV-2) is a β-coronavirus, enveloped RNA virus, belonging to the *Sarbecovirus* subgenus and subfamily Orthocoronavirinae; SARS-CoV-2 causes coronavirus disease 2019 (COVID-19). It emerged in Wuhan China and spread throughout China and globally through human mobility. Bats are believed to be the natural host of SARS-CoV-2 while zoonotic transfer and spillover through mammals such as ferrets, pangolins and mink are thought to be potential intermediate hosts to infect humans [1,2,3,4]. The World Health Organization (WHO) declared COVID-19 a Public Health Emergency of International Concern (PHEIC) [1] on 30 January 2020.

Globally, as of 19 November 2023, the COVID-19 pandemic affected over 772 million people in 229 countries and territories with over 7 million deaths [5]. Although the WHO declared COVID-19 no longer a PHEIC on 5 May 2023, it continues to constitute a public health threat monitored and addressed at the country level [6]. The potential spread of the disease, and especially new variants, remains a challenge to public health surveillance systems [7]. The WHO was tracking three variants of interest (XBB.1.5, XBB.1.16 and EG.5) and seven variants under monitoring (BA.275, BA.286, CH.1.1, XBB, XBB.1.9.1, XBB.1.9.2 and XBB.2.3) as of 24 September 2023 [8].

Many studies have shown the potential of communicable diseases to spread from one country to another through airports, ports and ground crossings [9,10]. To help prevent this spread, countries are required under articles 19, 20 and 21 of WHO International Health Regulation of 2005 to develop core capacities at designated POEs for routine times and during emergencies. These requirements highlight the importance of POEs in mitigating the importation and exportation of infectious diseases. During the COVID-19 pandemic, enhanced health screenings at POEs were among the public health measures implemented to detect, delay or prevent further transmission of COVID-19 early on.

Enhanced public health screenings at POEs may include requiring travelers to fill out a risk assessment questionnaire either before or upon arrival or before departure, observing for signs and symptoms of communicable illness, measuring body temperature, or conducting rapid testing. With evolving COVID-19 epidemiology and transmission dynamics including emerging new variants and increased coverage of COVID-19 vaccination, the WHO recommended thorough, systematic and regular risk assessments to inform the introduction, adjustment and discontinuation of risk mitigation measures in the context of international travel [11]. This risk-based approach involved considering different factors such as local epidemiology in departure and destination countries, the volume of travelers between countries, existing bilateral and multilateral agreements between countries to facilitate free movement, the capacity to detect and care for cases and their contacts, public health and social measures implemented to control the spread of COVID-19 in departure and destination countries, and contextual factors such as the economic impact, human rights impact and feasibility of applying measures [11]. As the COVID-19 pandemic continued, countries adjusted their POE screening measures based on response experiences, WHO COVID-19 travel advices and the emergence of new scientific ways of addressing the epidemic.

Several studies found varying results after evaluating the effectiveness of POE screening measures in the detection and containment of communicable diseases [10,12]. Some studies found POE screening to be effective in delaying cross-border spread and others showed less effectiveness in detecting febrile travelers. Some infectious diseases with a long incubation period such as Ebola and the limited accuracy of temperature-measuring devices were among the factors hindering the effectiveness of POE screenings [13,14].

The present systematic review seeks to complement the existing body of knowledge on the effectiveness of entry and exit health screenings at POEs to detect and contain COVID-19. It analyzes the existing body of knowledge for updated screening guidelines, screening effectiveness and temperature-measuring devices used and the potential gaps for further studies.

## 2. Materials and Methods

### 2.1. Eligibility Criteria

This study followed the PICO framework whereby the eligible population was international travelers, and the intervention was entry and exit health screenings at POEs. The comparator was other interventions implemented outside the POE settings such as contact tracing, surveillance through Integrated Disease Surveillance and Response (IDSR), quarantine and health facility reporting. The outcomes included detection of COVID-19 among travelers at POEs and the delay and containment of COVID-19 transmission between countries. The systematic review protocol was prepared and published under PROSPERO guidelines (available online: https://www.crd.york.ac.uk/prospero/display_record.php?ID=CRD42022336922, accessed on 8 June 2022).

### 2.2. Inclusion Criteria

The review included observational studies (cohort studies, case–control studies and cross-sectional), mathematical modeling and computational studies published worldwide in the English language in peer-reviewed journals and the gray literature between 2019 and 2022. These studies had to include an analysis of entry or exit screening practices at international airports, ports or ground crossings.

### 2.3. Exclusion Criteria

All reviews, editorials, full-text papers that were not available, studies involving migrant/asylum seekers and responses to COVID-19 in a particular conveyance which was part of the national response were excluded from the list. Publications reporting screening of other infectious diseases apart from COVID-19 such as TB, HIV and hepatitis were excluded.

### 2.4. Data Sources

The systematic search accessed many international databases for primary data sources including Scopus, PubMed, Web of Science, Global Health, CINAHL and Embase. Additional resources were gathered from Google Scholar and the gray literature was added from international organizations such as the WHO’s Institutional Repository for Information Sharing, the International Civil Aviation Organization (ICAO), International Maritime Organization (IMO), East Africa Community (EAC), Southern African Development Community (SADC) and Common Market for Eastern and Southern Africa (COMESA). The search of references of included studies was conducted and included in the study if criteria were met.

### 2.5. Search Strategy

#### 2.5.1. Search Topics and Definitions

This review covers the following key topics:(a)Points of Entry: as defined under the IHR, 2005, as a passage for international entry or exit of travelers, baggage, cargo, containers, conveyances, goods and postal parcels as well as agencies and areas providing services to them on entry or exit;(b)Entry health screening: public health measures (such as temperature screening, visual inspection for signs of illness, exposure and travel history assessment and testing) implemented at POEs upon arrival with the purpose of identifying travelers infected with or exposed to COVID-19 to mitigate importation of COVID-19;(c)Exit health screening: public health measures (such as temperature screening, visual inspection for signs of illness, exposure and travel history assessment and testing) implemented at POEs before departure with the purpose of identifying travelers infected or exposed to COVID-19 to prevent exportation of COVID-19 to other countries;(d)COVID-19 detection: confirmation of SARS-CoV-2 virus through accepted methods;(e)Travelers: passengers and crew under international voyage.

#### 2.5.2. Search Terms

The search terms were obtained through defining the synonyms of the terms, truncating and referencing from other similar published reviews. Table 1 below summarizes the terms used.

### 2.6. Data Extraction (Selection and Coding)

Studies were included if they evaluated health screening measures at POEs such as performance of fever screening devices, signs, symptom and exposure assessments, and rapid antigen test use among international travelers. These citations were exported to Rayyan (Rayyan System Inc, Cambridge, MA, USA) to screen based on the title and abstracts and to remove duplication. Rayyan is a free web and mobile app (http://rayyan.qcri.org, accessed on 20 June 2022) that expedites initial screenings. The software has been evaluated and found to be among the suitable tools for abstract and title screening [15,16]. Out of 33,744 documents retrieved, 109 were duplicates, which after removal, left a total of 33,635 subjected to level 1 screening. The title and abstract review excluded 33,199 documents, leaving 436 for full-text screening (level 2), based on the inclusion and exclusion criteria. Thirty-eight documents remained after the level 2 review. Searching the references of included studies led to including five more studies, making a total of 43 documents eligible for data extraction. Three independent reviewers (RK, EGK and EM) completed data extraction by using the data extraction protocol. Any disagreement among reviewers was resolved by discussion to reach consensus.

### 2.7. Quality of Included Articles

The quality of the included articles was assessed based on meeting the inclusion criteria.

### 2.8. Data Synthesis

All identified eligible studies were included in the synthesis. The review results were presented in Excel format, with columns representing the title of the article, author, year of publication, country where the study was conducted, screening strategy used, study design, objective, methodology, key findings, relevance, number of travelers screened, number of travelers found with COVID-19, type of POE (airport, port and ground crossing), performance of screening devices and synthesis of recommendations from an international organization. Health screening in this case refers to either symptom screening using risk assessment questionnaires, fever screening using infrared thermal scanners or illness screening using rapid antigen tests for COVID-19. The effectiveness of POE screening was assessed on the ability of these measures to detect COVID-19 cases, delay the onset of the COVID-19 epidemic and reduce transmission. After extracting data, the reviewers decided that a meta-analysis was not possible due to the nature of the data obtained. Research articles were heterogeneous and did not include all the information required including test statistics to enable meta-analysis.

## 3. Results

A total of 33,744 documents were initially retrieved through PubMed, 1680 (5% of 33,744); Scopus, 21,525 (63.8%); Embase, 1608 (4.8%); Google Scholar, 5310 (15.7%); Web of Science 1271 (3.8%); Global Health 2175 (6.4%); CINAHL 144 (0.4%); and national and international publications and the gray literature, 31 (0.1%). After completing level 1 and level 2 screening, the review analyzed 43 documents including 34 research papers and nine reports and guidelines published by the WHO and the United States Centers for Disease Control and Prevention (CDC) (Figure 1). Twenty articles were observational studies and 14 were prediction models. Most of the observational studies focused on airport screening practices, three reported on ground crossing and one paper described screening at a sea port. The results of this review are presented in various sections including the screening strategies used; effectiveness of POE screening in detection, delay and reduction in international transmission; effectiveness of temperature and symptom screening to detect COVID-19 among travelers at POEs; comparison between POE screening and other public health measures in detection and containment of COVID-19; and published recommendations from international organizations on how screening for COVID-19 should be carried out.

### 3.1. Screening Strategies Used at Points of Entry

Twenty observational studies reported on screening strategies used at POEs for the detection of COVID-19, as shown in Table 2. Testing of travelers was reported by 14 of the included studies [17,18,19,20,21,22,23,24,25,26,27,28,29,30], Sign and symptom assessment, especially loss of taste or smell, myalgia, cough, shortness of breath, self-reported fever, chills and vomiting, were reported by 10 out of 20 studies [20,21,27,28,30,31,32,33,34,35], followed by body temperature checks in seven studies [20,24,31,33,34,35,36] and exposure assessment by six other studies [20,21,27,30,32,35]. Three studies reported on the daily telephone calls of travelers after arrival for 14 days [20], checking vaccination status [22] and inspection of RT-PCR test certificates [27]. Table 2 below summarizes the screening strategies at POEs.

### 3.2. Effectiveness of POE Screening in the Detection, Delay and Reduction in Transmission of COVID-19

#### 3.2.1. Predictive Models on the Ability of POEs in the Detection, Delay and Reduction in Transmission of COVID-19

Fourteen models (Table 3a,b) predicted the effectiveness of POE screenings in the detection, delay and reduction in transmission risk. Five models [37,38,39,40,41] predicted the detection capacity and nine models [42,43,44,45,46,47,48,49,50] predicted the effectiveness of POE screenings in the delay and reduction in transmission of COVID-19 in unaffected communities.

Quilty, Clifford [37] evaluated the effectiveness of airport screenings by simulating 100 COVID-19-infected travelers expecting to transmit the disease to another region. Syndromic exit screening detected 44% of the infected travelers, entry screening detected 9% and 46% was not detected. Conversely, Bays, Bennett [38] showed that POE screening has a low detection rate and can only detect a maximum of 8.8% of travelers infected with COVID-19. The detection of COVID-19 is estimated at 94% if arrival screening (testing) is incorporated within eight days of isolation [41]. Similarly, entry and exit testing by RT-PCR combined with quarantine at 7, 14 and 21 days can detect COVID-19 cases at 95.1%, 98.5% and 99.6%, respectively, provided the source population prevalence is estimated at 0.1% to 2% per million travelers [39]. Conversely, another study [40] estimated the arrival and departure symptom screening to be able to detect 0.3% of infected travelers in the growing epidemic (5% subclinical) and arrival screening detecting one third (17–53%) with both entry and exit screening detecting nearly half of infected travelers in the stable epidemic (25% becoming subclinical).

In cases of delay of the epidemic onset, Clifford, Pearson [42] showed that syndromic entry and exit screening combined with sensitization can delay the local outbreak of COVID-19 in unaffected areas by 8 days. On the other hand, using the stochastic model, Nuckchady [43] predicted that assuming one infected person enters the country per day without the implementation of entry screening, the epidemic will occur within six days while implementing POE screening, with a sensitivity of 64%, and 100% will delay the onset of the COVID-19 epidemic for 10 and 20 days, respectively. Another study by Mandal, Bhatnagar [44] supported these findings, indicating that POE screening measures can delay the average time to epidemic for 20 days provided that the POE screening measures involve the diagnosis of 90% of asymptomatic travelers at a relative infectiousness of 0.1 between the asymptomatic and symptomatic and a reproductive number of 2. Hossain, Junus [47] modeled the effect of border control and quarantine in the importation of cases from Wuhan, China, to other 10 cities in China. It was observed that at reproductive numbers (Ros) of 1.4, 1.68 and 2.92, border control measures could reduce the arrival time to outbreak (cases reaching above 8) for an average of 32.5, 20 and 10 days, respectively. Another study assessed the impacts of testing, quarantine and symptom monitoring as travel-related strategies to reduce transmission [45]. The study found that symptom screening and immediate isolation can reduce the risk of disease transmission during traveling by 30–35% and testing all asymptomatic travelers during exit can reduce the risk of exportation by 44–72%.

Similarly, Dickens, Koo [46] created a simple model to estimate active case exportation risk from 153 countries with recorded COVID-19 cases and deaths. The study evaluated the impact of implementing different POE screening-related measures as opposed to no entry screening. Six strategies were evaluated including no screening, testing all travelers and isolating positives until they tested negative, with a retest at day 7 of isolation. The third strategy was the same as strategy two but travelers were allowed entry if they tested negative after a 14-day period. Strategy four involved quarantining all travelers for seven days before allowing entry. Strategy five required all travelers to quarantine for 14 days before they were allowed entry. Finally, strategy six included testing all travelers and preventing entry for those who tested positive. Countries implementing these strategies reduced their COVID-19 importation risk relatively to strategy one, on average, by 90.2%, 91.7%, 55.4%, 91.2% and 77.2%, respectively.

Steyn, Lustig [48] reported on the effect of combining border strategies in the reduction of transmission of COVID-19 among vaccinated and non-vaccinated travelers. Pre-departure PCR testing 3 days before departure and two tests post-arrival (days 0 and 4), combined with seven days of quarantine in Managed Isolation Quarantine Facilities (MIQF), reduced the transmission risk by 0.35% and 0.18% among non-vaccinated and vaccinated travelers, respectively. The risk of transmission was reduced if quarantine lengths were extended to 14 days in MIQ plus two PCR tests. Also, another study [49] compared different quarantine options combined with on-arrival testing on reduction in risk of infectious COVID-19 into the community. It was observed that testing on arrival alone could reduce the risk of sharing the disease to the community by 40–42%, while combining it with 21 days of quarantine, reduced the risk to almost zero. Similarly, Kiang [50] showed that antigen testing on the day of travel was associated with a 32% reduction in infectious days and 86% detection of active cases of COVID-19.

#### 3.2.2. Observational Studies on the Ability of POEs to Detect COVID-19 Cases

##### Confirmed Cases Detected through POE Screening at Points of Entry

Fourteen observational studies reported on the effectiveness of POE health screening in the detection of COVID-19, of which 10 studies were conducted at international airports [21,22,26,28,29,30,31,33,35,36], three at ground crossings [24,25,27] and one at a sea port [18], as shown in Table 4a,b. At the international airports, eight studies [21,22,26,28,29,30,31,35] reported the detection of suspected and confirmed cases of COVID-19 through POE screenings. Two studies reported zero confirmed cases out of the suspected cases [34,37]. At the ground crossings, all three studies [24,25,27] reported the detection of COVID-19 cases through POE screenings, similarly to one study conducted at an Italian sea port, where rapid testing detected 212 confirmed cases out of 272 suspected.

##### Comparison between COVID-19 Cases Detected through POE Screening and Other Public Health Interventions

Three studies reported on the comparison between POE screening and other non-POE-based public health measures such as Integrated Disease Surveillance and Response (IDSR), contact tracing, health facility reporting and quarantine [20,34,36]. In Taiwan, POE screening detected 32.7% of the imported cases as opposed to 27.7%, 16.2% and 23.4% detected through quarantine, contact tracing and health facility reports, respectively [34]. These findings were supported by a study in Zambia that analyzed the first 100 cases imported and found 35% were detected through POE surveillance [20], while the rest were identified through contact tracing, health care worker testing, health facility inpatient screening, community-based screening and calls from the public to a national hotline. Conversely, Aroskar, Sahu [36] found POE screening detecting zero COVID-19 cases out of 165,882 screened as compared to 49 cases detected through the Integrated Disease Surveillance Program (IDSP).

##### Detection of COVID-19 Cases through POE Testing Strategy and Genetic Sequencing

Ten observational studies reported on arrival testing for COVID-19 among international travelers [17,18,21,23,25,26,27,28,35,36], as shown in Table 5. A combination of molecular testing in the screening of travelers at POEs has been reported with positive outcomes, with one prospective cohort study reporting that out of 248 cases imported in the country, 167 (67%) were detected during entry screening for travelers entering Toronto, Canada, between September and October 2020 [21]. Also, in Venezuela, at a Maiquetia airport, molecular testing involving the sequencing of 256 samples obtained from tested travelers at POEs led to the detection of Omicron B.1.1, a variant of concern (VOC) [23]. Also, in Japan, at four airport quarantine stations (Narita, Nagoya, Kansai and Hanada), out of 168,061 travelers tested, 782 were confirmed with COVID-19, and further genomic sequencing of 129 samples identified lineage from Philippines, Pakistan and Brazil. The author recommended having a testing strategy which combines with genomic surveillance in order to prevent and monitor COVID-19 [26].

In Nepal, POE screenings were conducted at 13 designated borders (Pulghat, Jhulaghat, Gadachhauki, Tri Nagar, Jamunaha, Krishnanagar, Belahiya, Inarwa, Gaur, Bhittamod, Rani, Kakarvitta and Pashupatinagar) in response to a threat of a delta variant in India [27]. Rapid testing using a lateral flow antigen (LFA) detected 3907 cases of COVID-19 out 69,886 tested (travelers without an RT-PCR certificate or displaying signs and symptoms during primary screening) [27].

Similarly, in Pakistan, travelers entering the country through seven international airports from the UK and South Africa were subjected to rapid antigen testing and RT-PCR. This measure detected 243 cases of COVID-19 out of 74,833 travelers tested [35]. On the same note, 557 cases of COVID-19 were detected out of 155,087 screened at big airports in Japan between August and October 2020 using quantitative antigen testing, while 9873 COVID-19 cases were detected at the Ibrahim Al-Khalil border in Iraq between 21 August 2020 and 21 August 2021 out of 1,082,074 screened using RT-PCR [25,28].

Italy tested 38,282 passengers entering through a port using a rapid test, and among 272 positives, 212 were confirmed by qRT-PCR, with a detection rate of 77.9% [17]. Similarly, Yokota, Shane [18] reported that during the COVID-19 pandemic, a two-step screening strategy (rapid test followed by nucleic acid amplification tests (NAATs) only for intermediate-range antigen concentrations) was conducted at Japan’s three international airports. This approach was efficient and effective in detecting COVID-19 among international travelers. The authors concluded that there is a need for point-of-care testing at POEs as an initial screening process using quantitative rapid testing; screening measures can reserve NAAT testing for indeterminate tests. Colavita, Vairo [19] reported that a rapid test algorithm used in Italy identified 1176 COVID-19-positive travelers out of 73,643 screened. Out of the positive samples, 40.5% were confirmed by RT-PCR. The author concluded that rapid antigen testing is an important public health measure especially in settings with limited molecular tests such as at POEs.

##### Detection of COVID-19 Cases through POE Symptom and Temperature Screening Strategy

Eight studies reported on the effectiveness of symptom and temperature screening for the detection of COVID-19 [24,30,31,32,33,34,51], as shown in Table 6. A study conducted in China analyzed the characteristics of 1610 imported cases from 49 countries to 27 provinces and showed that 19.6% of the cases displayed signs and symptoms on the entry date. This finding highlights the importance of POE screening measures [51]. In Taiwan, Liu, Chen [34] indicated that body temperature and symptom screening at airports identified 32.7% of the 320- foreign imported cases in Taiwan during the period of 21 January to 6 April 2020. Similarly, another study [32] analyzed the epidemiological investigation forms and results of PCR testing for 11,074 arrivals at Incheon International Airport between 11 March and 30 April 2020 to determine the association between symptom screening and detection of COVID-19. The study found that 388 confirmed COVID-19 cases had reported a loss of smell and taste, self-reported fever, chills, a cough and vomiting. Additionally, requiring travelers to report their exposure to a confirmed case increased the detection level. Also, thermal screening at the Mutukula border in Uganda detected 57.8% with COVID-19 out of the thermally suspected travelers and contributed to 10% detection of all imported cases [24].

Conversely, in the USA, Dollard, Griffin [30] reported on the symptoms and exposure screening at 15 designated international airports. A total of 766,044 people were screened and 298 met the case definition, of which 35 were tested and nine were positive for COVID-19 [30]. The author noted that SARS-CoV-2 infection and transmission can occur in the absence of symptoms and because the symptoms of COVID-19 are nonspecific, symptom-based screening programs are ineffective for case detection. Also, Sujatha, Krishnankutty [31] showed that before the lockdown at Thiruvananthapuram International Airport, out of 46,139 screened, symptom screening detected 297 and 23 were detected through thermal screening. However, COVID-19 was only confirmed in six (2%) of the 320 detected. On the other hand, during post-lockdown, out of 44,263 screened, 671 were identified through symptom screening and 12 (0.03%) through temperature scanning, and COVID-19 was confirmed in 45 (6.7%) travelers [30].

The authors concluded that although the symptom and thermal screening yield was low, it helped to identify travelers infected with COVID-19 and raise awareness about COVID-19 preventive measures. This study was supported by Aroskar, Sahu [36], who indicated that at Mumbai International Airport, out of 165,882 passengers screened by a thermal scanner when arriving from COVID-19-affected countries, only three suspected cases were detected and all were RT-PCR negative. Even though the thermal screening may have had a low detection rate, Pasi, Gaikwad [33] concluded that collecting self-declaration forms provides a chance for travelers to declare their symptoms, sensitizes passengers and assists in initiating contact tracing.

#### 3.2.3. Guidelines and Documents Published by the WHO and CDC to Guide Screening for COVID-19 at POE

Eight publications were made by WHO and CDC from time to time to guide implementation of enhanced health screening for COVID-19 at POE. Table 7 below summarizes the guidance and recommendations provided.

## 4. Discussion

This systematic review collated data on the effectiveness of POE-based health screening measures in the detection and containment of the international spread of COVID-19. The scientific literature included in this review exhibited considerable heterogeneity, encompassing various study designs, units of observation and computational studies. Despite this variability, the findings presented contentious perspectives on the effectiveness of POE health screening. Several included studies reported that POE health screening measures demonstrated efficacy in detecting significant COVID-19 cases, including variants of concern [17,21,22,24,25,26,27,28,29,30,32,35,37,38,39,40]. Additionally, these measures were associated with a delay in the onset of the epidemic [42,43,44,45] and reduction in transmission risk [45,46,48,49,50]. Notably, the methodologies employed in different countries exhibited considerable diversity, involving a range of screening methods such as testing [18,19,20,21,22,23,24,25,26,27,28,29,30,31]; sign and symptom assessment [20,21,27,28,30,31,32,33,34,35]; body temperature measurement [20,24,31,33,34,35,36]; exposure assessment [20,21,27,30,32,35]; daily telephone calls of travelers after arrival for 14 days [37]; checking vaccination status [22]; and inspection of RT-PCR test certificates [27].

Additionally, this review highlighted a crucial aspect of POE health screenings predicted to delay the onset of outbreak of COVID-19 by between 8 and 32.5 days depending on the sensitivity, reproductive number and screening method employed [42,43,44,47]. Further, this review predicted the reduction in COVID-19 transmission risk during traveling by 30–35% through symptom screening and immediate isolation and reduction in exportation risk by 44–72% through the testing of all asymptomatic travelers during exit screening [45].

Specifically focusing on molecular tests, including rapid tests and RT-PCR as screening measures at POEs, this review indicated a high detection level, ranging from 94% to 99.6%, especially when combined with isolation and quarantine [37,38,40,46,51]. Observational studies also contributed valuable insights, revealing that POE screening detected between 6% and 77.9% of COVID-19 cases [17,18,19,21,27]. Recent instances, such as the detection of the Omicron B.1.1, a variant of concern (VOC), in Venezuela at a Maiquetia airport [23] and the identification of COVID-19 lineages in Japan at four airport quarantine stations [26], underscore the practical significance of molecular testing at POEs. Predictive models from selected studies, like the one by Quilty, Clifford [37], emphasized the effectiveness of syndromic exit screening, though acknowledging potential missed cases. Similarly, some selected observational studies found sign and symptom screening combined with thermal screening detecting between 2% and 57.8% [24,30,31,32].

Comparative effectiveness analyses suggested that POE screening outperformed other interventions such as contact tracing, Integrated Disease Surveillance and Response, quarantine and health facility reporting [20,59]. Notably, international organizations have continually published guidelines, recommending a risk-based approach over universal testing for all travelers [11,53,54,57]. COVID-19 vaccination was not a recommended mandatory requirement by the WHO for both entry and exit based on vaccination distribution inequalities among member states [54].

The findings summarized in this review underscore the crucial role of POE health screening in the early detection, delay and containment of the COVID-19 pandemic. Maximum benefit is derived when screening measures involve a combination of sign and symptom screening, temperature screening, testing and, when feasible, quarantine of exposed travelers. The considerable variation in detection levels may be attributed to differences in the sensitivity and specificity of screening devices, evolving border health policies and changing disease transmission dynamics.

These findings are consistent with other studies, including reviews. A similar review conducted to assess the travel-related control measures to contain the COVID-19 pandemic found health screening to be capable of delaying COVID-19 by between 1 and 183 days and capable of detecting the disease in between 10% and 53% [60]. Similarly, another review indicated that POE-based measures detected only half of the cases [10]. Other reviews indicated that thermal screening using non-contact thermal meters was ineffective in limiting the spread of the COVID-19 pandemic [37,61], similar to what was found in this study. This review found better detection when screening strategies involved a combination of testing, similar to the results of other review articles [60]. Furthermore, Gunaratnam, Tobin [62] found that during the 2009 H1N1 influenza pandemic, 5845 travelers screened at Sydney International Airport, New South Wales, Australia, were identified as febrile out of 625,147 total screened during that period. Three travelers were confirmed with H1N1 (detection rate of 0.05 per 10,000 screened). The author estimated that during the same period, 45 cases of H1N1 passed through that airport, thus estimating the sensitivity values to be 6.67%.

The present review found POE screening more effective than other interventions in preventing the importation of COVID-19 cases, similar to another study by Zhang, Yang [63], who found POE screening to be the second most effective intervention in detecting imported cases of H1N1 compared to influenza-like illness screening in hospitals, medical follow-up of travelers from overseas and quarantined close contacts. On the contrary, in Australia, most H1N1 cases were detected in emergency departments [62]. The variation in screening performance observed in this review may be due to a lack of uniformity in the application of screening strategies as well as screening capacity-related challenges. In India, at Jaipur International Airport, Neha, Joshi [64] reported poor cooperation among passengers, masking symptoms, apprehension and inadequate human resources as barriers to POE screening effectiveness. Similarly, other studies have reported inadequate personal protective equipment and supplies, insufficient screening infrastructure, inadequately trained staff and the existence of many informal entry points, lack of training among staff, missing information in health alert cards, lack of harmonized traveler screening measures, poor intersectoral coordination, lack of transport, absence of an public health emergency plan, and inadequate financial resources as potential challenges affecting POE health screening [65,66,67,68].

The main limitation of this study is that many included papers reported POE screening for airports and ground crossings with limited studies for ports. The reported findings might have underreported what is being implemented at ports. Also, it is impossible to know how many travelers had transmissible COVID-19 during their travel through a POE and thus, it was difficult to judge whether they missed the cases or they detected the actual number. Further, risk of importation and exportation of COVID-19 varied throughout the pandemic based on a variety of border health policies implemented around the world which impacted screening measures and when, where and how individuals were traveling. Moreover, future reviews should consider conducting a meta-analysis which was not performed due to heterogeneousness of the studies obtained.

## 5. Conclusions

This review found that enhanced health screening involving multiple strategies such as temperature checks, health declarations, visual inspection, exposure assessment and testing is effective in the detection of COVID-19 cases and delayed the onset of epidemic by between 7 and 32 days. We recommend a regular capacity and risk assessments at airports, ports and ground crossings and developing required core capacities for routines and during emergencies in line with IHR, 2005. These capacities will not only support COVID-19 detection and containment but also many other public health events with potential for international spread. We also recommend further studies to investigate the effectiveness of POE health screenings at ports and to evaluate the effectiveness of the available screening and preventive measures at POEs in the detection and containment of COVID-19.

## Figures and Tables

**Figure 1 ijerph-21-00410-f001:**
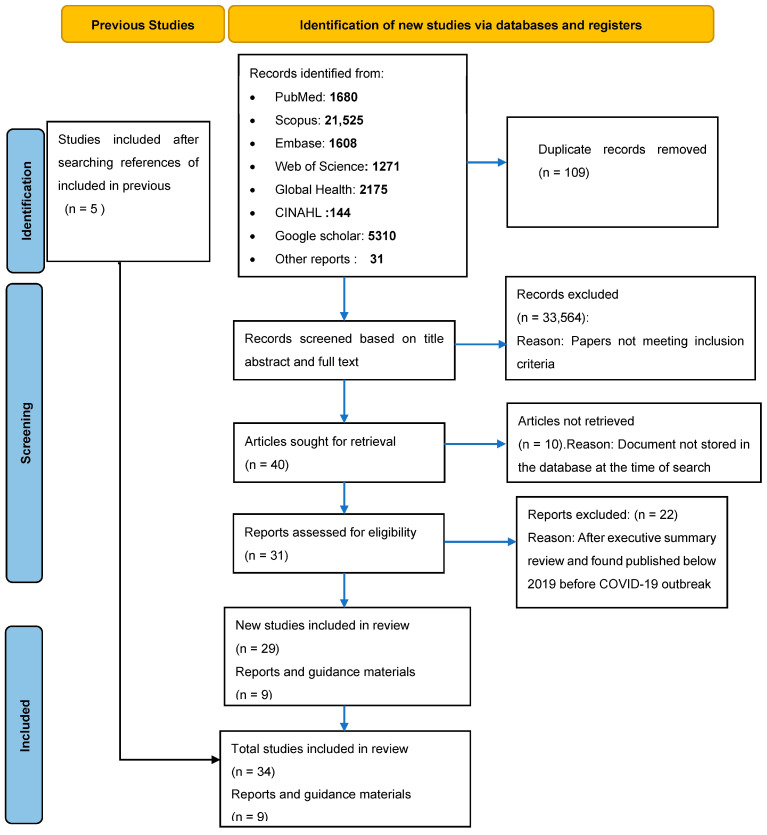
Prisma flow chart on the systematic review process.

**Table 1 ijerph-21-00410-t001:** Search terms.

Population	Intervention	Setting	Outcome
			COVID-19
			Coronavirus
travel*			S-COVID-19
passenger		point of entry	Coronavirus disease 2019
crew*	screen*	port of entry	SARS-CoV-2
driver*	measure*	depart*	Covid infection
conductor*	IR system*	arriv*	2019 Novel Coronavirus
truck driver*	thermal scan*	airport*	Covid disease
tourist*	non-contact thermometer*	point of departure	2019-nCoV
	non-contact infrared thermometer*	seaport*	Coronavirus Disease 2019
	questionnaire*	port*	COVID-19 Virus
	visual observation	entry point*	Virus Infection
	non-contact infrared camera*	quarantine station*	Severe Acute Respiratory Syndrome Corona virus2
	infrared thermo*	railway station*	SARS Coronavirus2
	surveillance form*	border*	Virus Disease
	surveillance	ground crossing*	Pandemic
	sign*	exit point	Epidemic
	symptom*		Wuhan*

Search string: (travel* OR passenger OR crew* OR driver* OR conductor* OR “truck driver*” OR tourist*) AND (screen* OR measure* OR “IR system*” OR “thermal scan*” OR “non-contact thermometer*” OR “non-contact infrared thermometer*” OR questionnaire* OR “visual observation” OR “non-contact infrared camera*” OR “infrared thermo*” OR “surveillance form*” OR surveillance OR sign* OR symptom*) AND (“point of entry” OR “port of entry” OR depart* OR arriv* OR “point of departure” OR airport* OR seaport* OR port* OR “entry point*” OR “quarantine station*” OR “railway station*” OR border* OR “ground crossing*” OR “exit point”) AND (“COVID-19” OR Coronavirus OR “S-COVID-19” OR “Coronavirus disease 2019” OR “SARS-CoV-2” OR “Covid infection” OR “2019 Novel Coronavirus” OR “Covid disease” OR “2019-nCoV” OR “COVID-19 Virus” OR “Virus Infection” OR “Severe Acute Respiratory Syndrome Corona virus2” OR “SARS Coronavirus2” OR “Virus Disease” OR Pandemic OR Epidemic OR Outbreak OR Wuhan*).

**Table 2 ijerph-21-00410-t002:** Strategies used for screening at POEs.

Country/POE	Exposure Assessment	Sign and Symptom Assessment/Self-Declaration	Temperature Measurement	Testing	Other	Ref.
India: Thiruvananthapuram International Airport	NR	Yes	Yes	NR	NR	[31]
Republic of Korea: Incheon International Airport	Yes	Yes	NR	NR	NR	[32]
Italy: Sea port	NR	NR	NR	Yes	NR	[17]
India: Cochin, Delhi Kolkata and Mumbai airports	NR	Yes	Yes	NR	NR	[33]
India: Mumbai International Airport	NR	NR	Yes	NR	NR	[36]
Japan: Three international airports	NR	NR	NR	Yes	NR	[18]
Italy: international airports in Rome and port of Civitavecchia	NR	NR	NR	Yes	NR	[19]
Zambia: Points of entry surveillance	Yes	Yes	Yes	Yes for symptomatic	Daily telephone call for 14 days	[20]
Canada: Toronto’s Pearson International Airport	Yes	Yes	NR	Yes	NR	[21]
Taiwan	NR	Yes	Yes	NR	NR	[34]
Quatar: Hamand International Airport	NR	NR	NR	Yes	Vaccination status of travelers	[22]
Venezuela: Maiquetia airport	NR	NR	NR	Testing and sequencing	NR	[23]
Uganda: Mutukula border	NR	NR	Yes	Yes	NR	[24]
Iraq: Ibrahim Al-Khalil border	NR	NR	NR	Yes	NR	[25]
Japan: Narita, Hanada, Nagoya and Kansai airports	NR	NR	NR	Testing and sequencing	NR	[26]
Pakistan: Seven airports	Yes	Yes	Yes	NR	NR	[35]
Nepal: 13 designated borders	Yes	Yes	NR	Yes for suspect travelers or without negative RT-PCR	RT-PCR test certificates	[27]
Japan: Five major airports	NR	Yes	NR	Yes	NR	[28]
Alaska: 10 airports	NR	NR	NR	Yes	NR	[29]
USA: 15 designated airports	Yes	Yes	NR	Yes for suspects	NR	[30]

NR = Not reported.

**Table 3 ijerph-21-00410-t003:** (**a**) Models predicting the effectiveness of POE screening in the detection of COVID-19 cases. (**b**) Models predicting the effectiveness of POE screening in the delay in COVID-19 outbreak and reduction in international spread in unaffected countries/communities.

(a)				
Title of Study	Model Type	Detection of Positive COVID-19 Cases		Ref.
		Exit Health Screening	Entry Health Screening	
1. Effectiveness of airport screening at detecting travelers infected with novel coronavirus (2019-nCoV)	Simulation	44%	9%	[37]
2. What effect might border screening have on preventing the importation of COVID-19 compared with other infections? A modeling study	Mathematical Modeling	NR	8.8%	[38]
3. What effect might border screening have on preventing importation of COVID-19 compared with other infections? Considering the additional effect of post-arrival isolation	Simulation		Single arrival test could detect 9% and combined with 8 days, isolation detection could reach 94%	[41]
4. Determining quarantine length and testing frequency for international border opening during the COVID-19 pandemic	Simulation	At a prevalence of 0.1% to 2% per million travelers, exit and entry testing by PCR combined with quarantine at 7, 14 and 21 days reduced missed cases at rates of 4.9%, 1.5% and 0.4%, respectively, and using rapid antigen testing, the reduction was at 3.6%, 2.8% and 0.7%, respectively.		[39]
5. Estimated effectiveness of symptom and risk screening to prevent the spread of COVID-19	Mathematical modeling	In the growing epidemic, under the assumption that 5% were subclinical, departure and arrival screenings detected 0.3% of infected travelers. While in the stable epidemic, with 25% being subclinical, arrival screenings alone detected one third (17–53%), and departure and arrival detected half (23–63%)		[40]
(**b**)				
**Title of Study**	**Model Type**	**Delay of COVID-19 Outbreaks and Reduction in Transmission Risk of Positive Cases**		**Ref.**
1. Effectiveness of interventions targeting air travelers for delaying local outbreaks of SARS-CoV-2	Stochastic model	POE entry and exit screenings, combined with traveler sensitization, delayed an outbreak by 8 days (50% interval: 3–14 days)		[42]
2. Impact of public health interventions on the COVID-19 epidemic: A stochastic model based on data from an African island	Stochastic model	A POE screening at 64% or 100% sensitivity delayed the onset of the COVID-19 epidemic for 10 to 20 days, respectively		[43]
3. Prudent public health intervention strategies to control the coronavirus disease 2019 transmission in India: A mathematical model-based approach	Mathematical model	If all symptomatic travelers were identified and 90% were asymptomatic, diagnosed POE screening could delay the epidemic by up to 20 days		[44]
4. Reducing travel-related SARS-CoV-2 transmission with layered mitigation measures: Symptom monitoring, quarantine and testing	Mathematical model	Symptom check at departure reduced risk of transmission for 30–35%, exit screening (testing) reduced the transmission risk by 44–72% and symptom evaluation reduced the risk of importation by 42–56%		[45]
5. Strategies at points of entry to reduce importation risk of COVID-19 cases and reopen travel	Simulation	Testing all travelers, isolating the positive and permitting entry after a negative test result at day 14 reduced importation risk by 91.7%		[46]
6. The effects of border control and quarantine measures on the spread of COVID-19	Mathematical model	At reproductive numbers (Ros) of 1.4, 1.68 and 2.92, border control could delay (above threshold of 8 cases in the community) the onset of outbreak for 32.5, 20 and 10 days, respectively		[47]
7. Effect of vaccination, border testing and quarantine requirements on the risk of COVID-19 in New Zealand: A modeling study	Mathematical model	Three-day pre-departure PCR test, followed by two PCR tests post-arrival (at 0 and 4 days) and managed isolation quarantine (MIQ) for seven days reduced COVID-19 transmission risk to 0.35 and 0.18 among non-vaccinated and vaccinated travelers, respectively, and a fourteen-day stay in MIQ with two PCR tests reduced the transmission potential to a negligible level		[48]
8. The differential importation risks of COVID-19 from inbound travelers and the feasibility of targeted travel controls: A case study in Hong Kong	Modeling study	On arrival, testing prevented an average of 40–42% of infectious travelers at the airport from mixing with the community		[49]
9. Routine asymptomatic testing strategies for airline travel during the COVID-19 pandemic: A simulation study	Simulation	Rapid antigen testing performed on day of departure was associated with reduction in infectious days by 32% and active infections of SARS-CoV-2 by 86%; when combined with a day 5 PCR test and 5 days of quarantine, infectious days were reduced by 70% and active infections, by 86%		[50]

NR = Not reported.

**Table 4 ijerph-21-00410-t004:** (**a**) Effectiveness of entry screenings at points in the detection of COVID-19 among travelers at POEs. (**b**) Effectiveness of entry screenings at points in the detection of COVID-19 cases at ground crossings and borders.

(a)							
Country	Title of Study	International Airport	Travelers Screened	Duration	Suspects Detected at Airport	Confirmed	Ref.
Kerala, India	Entry screening at airport as a COVID-19 surveillance tool (pre-lockdown)	Thiruvananthapuram	46,139	29 January to 24 March 2020	320	6 (1.9%)	[31]
Kerala, India	Entry screening at airport as a COVID-19 surveillance tool (post-lockdown)	Thiruvananthapuram	44,263	13 May to 31 July 2020	684	12 (1.8%)	[31]
India	Early detection of suspected cases of COVID-19: Role of thermal screening at international airports in India	Cochin, Delhi, Kolkata and Mumbai	1,587,034	17 January to 30 September 2020	151	0 (0%)	[33]
India	Evaluation of point of entry surveillance for COVID-19 at Mumbai International Airport	Mumbai	165,882	1 to 22 March 2020	3	0 (0%)	[36]
Canada	COVID-19 international border surveillance at Toronto’s Pearson Airport: A cohort study	Pearson	16,361	September to October 2020	NR	167	[21]
Quatar	Associations of vaccination and of prior infection with positive PCR test results for SARS-CoV-2 in airline passengers arriving in Qatar	Hamad	247,091	18–26 February April 2021	NR	8319	[22]
Japan	COVID-19 genome surveillance at international airport quarantine stations in Japan	Narita, Hanada, Nagoya and Kansai	168,061	March to 1 September 2020	NR	782	[26]
Pakistan	Descriptive analysis of health screening for COVID-19 at points of entry in Pakistan according to the Centers for Disease Control and Prevention guidelines	Seven international airports	361,737	February 2020 to March 2021	375	NR	[35]
Pakistan	Descriptive analysis of health screening for COVID-19 at points of entry in Pakistan according to the Centers for Disease Control and Prevention guidelines	Seven international airports	74,833	February 2020 to March 2021	NR	243	[35]
Japan	Epidemiology and risk of coronavirus disease 2019 among travelers at airport and port quarantine stations across Japan: A nationwide descriptive analysis and an individually matched case–control study	Five major international airports	155,087	August to October 2020	558	0.35%	[28]
Alaska	Airport traveler testing program for SARS-CoV-2—Alaska, June–November 2020	10 participating airports	111,370	6 June to 14 November 2020	951	0.8%	[29]
USA	risk assessment and management of COVID-19 Among travelers arriving at designated U.S. airports, 17 January–13 September 2020	15 designated international airports	766,044	17 January to 13 September 2020	298	9 (0.001%)	[30]
(**b**)							
**Country**	**Title of Study**	**International** **Border/Sea Port**	**Travelers Screened**	**Duration**	**Suspects Detected at POE**	**Confirmed**	**Ref.**
Italia	Prevention of the spread of SARS-CoV-2 by rapid antigenic tests on the passengers entering an Italian seaport	Sea port	38,282	21 August to 27 September 2020	272	212 (77.9%)	[17]
Nepal	COVID-19 amongst travelers at points of entry in Nepal: Screening, testing, diagnosis and isolation practices	13 designated borders	337,338	March to July 2021	69,886	3907 (6%)	[27]
Iraq	SARS-CoV-2 and RT-PCR testing in travelers: Results of a cross-sectional study of travelers at Iraq’s international borders	Ibrahim Al-Khalil border	1,082,074	21 August 2020 to 21 August 2021	9873	0.9%	[25]
Uganda	Effectiveness of thermal screening in detection of COVID-19 among truck drivers at Mutukula land point of entry, Uganda	Mutukula border	7181	15 May to 30 July 2020	83 suspected by thermal scanner	48 (57.8%), actual confirmed cases by Xpert Xpress SARS-CoV-2 assay, 483; thus, detection rate of 10%	[24]

NR = Not reported.

**Table 5 ijerph-21-00410-t005:** Detection of COVID-19 at points of entry through rapid testing.

Country, POE	Detection Strategy	Travelers Screened	Suspected/Total Imported Cases	Confirmed at POE	Proportional COVID-19 Detected at POE	Ref.
Italy	Rapid test	38,282	272	212	77.9%	[18]
Italy	Rapid test	73,643	1173	476	40.5	[19]
Toronto, Canada	Rapid test	16,361	248 imported	167	67.30%	[21]
Japan	Rapid test	88,924	513	34	6.60%	[18]
Venezuela, Maiquetia airport	Molecular test with sequencing	256 samples of travelers	NA	Omicron B.1.1 (VOC)	NA	[23]
Iraq, Ibrahim Al-Khalil border	RT-PCR	1,082,074	NR	9873	0.9% of those screened	[25]
Japan	Molecular test with sequencing	168,061	NR	782	129 samples sequenced identified lineages from three foreign countries	[26]
Pakistan	RT-PCR and rapid antigen test	74,833		243		[35]
Japan	Antigen test	155,087		558		[28]
Nepal	Lateral flow antigen test	69,886		3907	6% positivity	[27]

NA = Not applicable; NR = Not reported

**Table 6 ijerph-21-00410-t006:** Detection of COVID-19 cases at points of entry through sign, symptom and temperature assessments.

Country	Screening Strategy	Number Screened	Suspected	Confirmed	Proportion Confirmed among Suspects	Proportion Detected at POE among Imported	Ref.
China	Signs and symptoms	5,291,039	NR	1610	NA	315 (19.6%)	[51]
Taiwan	Signs, symptoms and temperature checks	NR	NR	320	NA	105 (32.7%)	[34]
Republic of Korea	Signs, symptoms and temperature checks	348,753	11,074	388	3.5%	No data on total imported cases	[32]
India—Thiruvananthapuram airport	Signs, symptoms and temperature checks	46,139	320	6	2%	No data on total imported cases	[31]
India—Thiruvananthapuram airport	Signs, symptoms and temperature checks	44,263	683	45	6%	No data on total imported cases	[31]
Uganda, Mutukula border	Temperature checks	7181	83	48	57.8%	488 (10%)	[24]
USA airports	Exposure and symptom screening	766,044	298	9	3%	No data on total imported cases	[30]
India airports	Temperature checks	1,593,861	151	NR	NR	NR	[33]
India, Mumbai International Airport	Temperature checks	165,882	3	0	0	NR	[36]

NA = Not applicable; NR = Not reported

**Table 7 ijerph-21-00410-t007:** Published guidelines and recommendations from international organizations about screening for COVID-19 at POEs.

Publication	Title	Key Recommendations/Findings
WHO [52]	Technical considerations for implementing a risk-based approach to international travel in the context of COVID-19: interim guidance: Policy considerations for implementing a risk-based approach to international travel in the context of COVID-19, 2 July 2021	During the COVID-19 pandemic humanitarian missions, travel of essential personnel, repatriations and cargo transport of essential supplies should be prioritized Introduction of risk mitigation measures aiming to reduce travel-associated exportation, importation and onward transmission of SARS-CoV-2 should be based on thorough risk assessments conducted systematically and routinely The application of a precautionary approach is warranted in the presence of scientific uncertainties such as the emergence of variants of concern (VOCs) or variants of interest (VOIs) Proof of COVID-19 vaccination should not be required as a condition of entry to or exit from a country Testing or quarantine as a condition for entry may consider exempting vaccinated or previous infected travelers Adherence to personal protective measures such as mask use and physical distancing must continue to be respected by all international travelers, both while on board conveyances and at POEs International travelers should not be considered by default as suspected COVID-19 cases or contacts or as a priority group for testing
WHO [11]	Evidence reviews—Public health measures in the aviation sector in the context of COVID-19: Quarantine and isolation	Through a systematic review, the WHO concluded that the evidence on usefulness of quarantine to prevent transmission of SARS-CoV-2 is of low to very low certainty at best and based on a limited number of modeling studies and a few observational series conducted up to 13 November 2020. Therefore, the implementation of international travel and health guidelines and isolation of symptomatic and/or SARS-CoV-2 test-positive travelers were endorsed as a response strategy to the COVID-19 pandemic.
WHO [53]	Operational framework for international travel-related public health measures in the context of COVID-19	The following considerations should be taken onboard when deciding to implement international related public health measures: Recommend the use of risk-based, evidence-based, coherent, proportionate to the public health risk and unnecessary interference with international traffic and trade to prioritize groups for travel restriction; Develop a protocol in coordination with POE authorities for screening and identifying the required staff and resources to operate and encourage member states to share evaluation reports on the effectiveness of POE screening; Healthy travelers should not be designated as a priority group for SARS-CoV-2 testing, especially if testing resources are limited and high-risk groups such as vulnerable populations and health workers, including health workers in training and support services (such as laboratory and cleaning services), should be prioritized for testing; International contact tracing health declaration forms and the Passenger Locator Form (PLF) before departure are recommended through the use of the national IHR focal point and bilateral exchange of information and traveler-related data should be handled confidentially; Quarantine of travelers should be based on risk assessments and capacity of the country to implement, and travelers should not be charged for isolation or quarantine; Management of suspected cases shall entail detecting, interviewing and reporting alerts of ill travelers with suspected COVID-19 to local or national health authorities for isolation, initial case management and referral, And these measures should be included in POE public health contingency plans; Proof of COVID-19 vaccination in the context of international travel (digital or paper-based) is not recommended due to vaccine distribution inequalities;
WHO [54]	Policy considerations for implementing a risk-based approach to international travel in the context of COVID-19	Based on evolving evidence and the changing epidemiology of COVID-19, the WHO policy recommends the following for international travel-related public health measures: Member states should not require proof of COVD-19 vaccination as a mandatory condition for entry to or exit from a country; Consider a risk-based approach by lifting measures, such as testing and/or quarantine requirements, to individual travelers who were fully vaccinated, at least two weeks prior to traveling, with COVID-19 vaccines listed by the WHO for emergency use or who have had previous SARS-CoV-2 infection as confirmed by RT-PCR within the 6 months prior to traveling and are no longer infectious as per the WHO’s criteria for releasing COVID-19 patients from isolation; The use of serologic assays is not recommended to prove recovery status given the limitations that are outlined in the scientific brief “COVID-19 natural immunity”; Offer alternatives to travel for individuals who are unvaccinated or do not have proof of past infection, such as through the use of negative RT-PCR tests, or antigen detection rapid diagnostic tests (Ag-RDTs) that are listed by the WHO for emergency use or approved by other stringent regulatory authorities; Consider recording proof of COVID-19 vaccination in the International Certificate of Vaccination or Prophylaxis (ICVP) as stated in the WHO’s interim position paper, “Considerations regarding proof of COVID-19 vaccination for international travelers”; Observe dignity, human rights and fundamental freedoms of travelers during quarantine; Continue conducting regular and thorough risk assessments to update international travel-related measures as the situation evolves, particularly when VOIs and VOCs emerge
WHO [55]	Considerations for sharing information for international contact tracing in the context of COVID-19	IHR’s national focal point should be used for sharing contact information internationally: Information shared should be handled confidentially through encryption and password protection methods; Information can be shared in two ways involving initial information and additional information if requested
WHO [56]	Management of ill travelers at points of entry (international airports, seaports and ground crossings) in the context of COVID-19	The interim guidance provides for procedures for the detection and management of ill travelers suspected to have COVID-19 at POEs and on conveyances of all types specifically: Detection of ill travelers at POEs: staff should be trained on infection prevention and control (IPC) and detection should be performed through self-reporting by travelers on signs and symptoms, visual observation and temperature measurement; Interview of ill travelers: standard operating procedures (SOPs) should be available to staff to interview travelers for signs and symptoms, to record body temperature, and to document travel and recent health history; Reporting of alerts of ill travelers with suspected COVID-19: use of health section of aircraft general declaration and maritime declaration of health to reports alerts from aircrafts and marine vessels, respectively; Isolation, initial case management and referral of ill travelers with suspected COVID 19: adhere to IPC measures when handling infected travelers in isolation facilities
CDC [57]	Testing for SARS-CoV-2 infection at air, land and sea points of entry and complementary measures to limit international spread of COVID-19: Strategies for port health leaders outside the United States	Designing a testing strategy at POEs should consider the following: Throughput, performance, test availability, consequence management and characteristics of the POE; POE site-specific SARS-CoV-2 risk assessments; Antibody test not recommended at POEs as they do not detect active infection; When using antigen tests, consider how to implement a confirmatory test; Active or current infection status; Consider country context and types of POEs
CDC [58]	Tool to prioritize point of entry and point of control (POE/C) considerations for prioritizing points of entry and control for public health capacity building	Enhancing capacity building in the prevention of the transmission of communicable diseases through POEs (international) or points of control (domestic) can be prioritized based on the following criteria: Characterization of POE/C: based on security, infrastructure, staffing and whether the local community uses the POE/C for purposes other than travel, i.e., shopping; Traveler volume: number of people using a POE/C to enter or exit can elevate absolute risk for the international importation or exportation of communicable diseases; Connectivity to priority or high-risk populations or locations: travelers passing through a POE/C who come from or have connection to areas or populations affected by a communicable disease of public health concern are a major risk factor for the international importation or exportation of communicable diseases or domestic spread of disease between administrative areas; Ability of POE/C staff and infrastructure to manage sick travelers: capacity to identify and manage ill travelers is likely to mitigate the risk of importing or exporting communicable diseases through the POE/C; Strength of public health surveillance systems: a POE/C can serve as a public health surveillance site contributing to public health event detection along the border or at priority locations across the country, and POEs in areas with weaker community-based surveillance can be useful in the detection of ill travelers passing through these sites; Cross-border coordination: routine communication and information sharing on potential public health events between the local-level and cross-border counterparts and other public health stakeholders is critical to mitigating the spread of communicable diseases in border regions

## Data Availability

All data generated or analyzed during this study are included in this manuscript.

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
