# Peer review of "Effectiveness of Point of Entry Health Screening Measures among Travelers in the Detection and Containment of the International Spread of COVID-19: A Review of the Evidence"

_ijerph, 2024, doi:10.3390/ijerph21040410_

Round 1

Reviewer 1 Report (Previous Reviewer 1)

Comments and Suggestions for Authors

The paper now sounds good, except, a few issues as indicated in the attached file

Author Response

Dear Reviewer,

Kindly see our responses attached

Thanks

Reviewer 2 Report (Previous Reviewer 3)

Comments and Suggestions for Authors

Dear authors

All my previous comments, observations and recommendations, as well as suggestions to improve the manuscript and have the quality to be published were resolved and corrected. As far as I am concerned, it satisfies my requirements and I consider that it can follow its process according to the indications of the editorial committee of its well-known journal.

I mark some minor issues in the pdf attached  to materialize   improvements  for consideration for publication.

Author Response

Dear Reviewer,

Kindly receive our responses attached, we thank you for your valuable time and inputs provided

Kakulu

This manuscript is a resubmission of an earlier submission. The following is a list of the peer review reports and author responses from that submission.

Round 1

Reviewer 1 Report

Comments and Suggestions for Authors

Review report

EFFECTIVENESS OF POINT OF ENTRY HEALTH SCREENING MEASURES AMONG TRAVELERS IN THE DETECTION AND CONTAINMENT OF THE INTERNATIONAL SPREAD OF COVID-19: A REVIEW OF THE EVIDENCE

General observations

  • The paper describes an important aspect of international health especially at this era where international traveling is on increase day by day fueling transfer of communicable diseases from on continent to the other and from one country to the other. It highlights the key role of port health authorities in the prevention and control of communicable diseases from one country to the other with a focus on COVID-19. The paper adds a value on the implementation of IHR 2005, specifically on key core capacities. The readers will be interested because during COVID-19 there were stringent screening and other public health measures at POE across the world and travelers would wish to how much effective were the measures.
  • However,
    • the review has been so general, the ability to implement recommended core capacities at POE differ from one region to the other depending on the available resources. The findings could have been more instrumental had been that the review was carried out for a specific region, for example African countries or Asia etc. 
  • I was wondering if the findings can be generalized to other communicable diseases fueled by international travel such as SARS, Influenza and the like.  

_______________________________________________________________

Specific comments in the Table below

S/N

Line/section

Comments

Line 2

Since the review has considered other POE measures other than screening, you may consider to revise the title by omitting the word screening so you have all “health measures” included instead of health screen measures

That means the title could read “EFFECTIVENESS OF POINT OF ENTRY HEALTH MEASURES AMONG TRAVELERS IN THE DETECTION AND CONTAINMENT OF THE INTERNATIONAL SPREAD OF COVID-19: A REVIEW OF THE EVIDENCE”

Line 34

Begin with introducing what causes COVID-19 so that line 50 where you mentioned “SAR-COV2 virus” makes sense

Yes COVID-19 is familiar but again it is good idea you briefly explain the genesis of the disease in the introductory section

Line 40

The reader may be interested to know more about current new COVID-19 variants. I suggest you cite examples of current new variants of concern as reported by the WHO

Line 49

The sentence beginning with “Many countries….” do not sound better, its better to mention the total number of countries and provide a reference OR recast the statement

Line 51

With reference to line 49, countries activated enhanced screening, so the statement in line 51 …countries may…is contradicting. Just continue explaining what countries did as part of enhanced screening

Line 90

The list provided under exclusion are in fact already excluded with reference to inclusion criteria provided. You exclude what has been included. I suggest you restate the exclusion criteria from what has been included in the inclusion criteria

Line 132-133

Readers may be interested to know little more about Rayyan software. Provide more details or reference/citation for Rayyan software

Line 135-138

This may change subject to revised inclusion and exclusion criteria?

Refer comment on line 90. If you revise and restate the inclusion and exclusion criteria, this may change as well

Line 156

Insufficient explanation! Readers would like to know the specific reason for not conducting Meta- analysis as this was expected in our review

Line 404

From the findings, it seems there are no evidence that came through ground crossing as one of the POE. I addition, you may consider to add sea port as well since there are no findings indicating the screening practices

Line 411

The conclusion indicate that screening may not be a good way to detect communicable diseases at POE. This is worrisome because many governments especially in developing countries have heavily invested in POE screening. For this case I would suggest the recommendation focus on why screening is not doing well as expected? Is it related with insufficient capacity (human and technological)? What is the recommendation to IHR (2005) if the review is to be done? What should the guidelines on disease prevention at POE consider apart from normal screening? What should be enhanced? Should the governments continue to invest in POE screening? If not, what are the alternatives?

So instead of recommending further studies on the effectiveness of screening at ground crossing and sea port (where we do not expect much difference from airports or even worse) I suggest the recommendation focus on the above arguments.

Comments on the Quality of English Language

I am not an English speaker, however, based on my experience in scientific writing, the English presented in this paper has few typos and grammatic errors. 

Author Response

Dear Sir/Madam,

Greetings,

Thank you for your valuable comments,

Kindly see the attached our responses for your consideration.

We will be happy to clarify and matter as deemed necessary

Thanks,

Reviewer 2 Report

Comments and Suggestions for Authors

Keywords: Points of entry are synonyms of ports, airports, and borders, they cannot be put as separate entities in the keywords section

Introduction:

Lines 37 - 38: mention the date on which WHO declared COVID-19 as PHEIC

Line 49: Countries activated enhanced screening (choose one word, either activated or enhanced, but not both)

Line 50 SAR COV-2, should be SARS COV-2, again I should advise you to continue using COVID-19, as it's used throughout the manuscript.

Line 54: Kindly describe briefly the risk-based screening approach at POE as per WHO criteria

Line 58-59: Can you please explain more on the reasons why there is low evidence to support some of the public health measures applied at POE such as quarantine

Lines 62-65: Too long sentence, kindly shorten it into two sentences

Line 79: List other screening measures at POE that are considered as comparators - in a PICO platform

Lines 172-180: when you listed records identified, put a number of grey literature (22 records) so that you are able to get a total # of records to be 830

Line 138: says a total of 26 articles were eligible for data extraction, which is different from what you said in lines 199-203 that a total of 18 studies were included for data extraction, please harmonize these statistics for clarity.

Line 138: Add the word 'period' after incubation, so it reads: pathogens with a short incubation period.

Line 210: List a few examples of the signs and symptoms that were reported

Results section: In general; there is a lot of wording that needs to be removed and placed under the discussion section. I am of the opinion that the results section should carry only what you got from the extracted data analysis you had. For instance, lines 217 - 252, lines 269-335 should go to the discussion section.

Discussion section: This is where you need to have more explanations of your key findings and compare them with what other studies have reported in a similar context.

Comments on the Quality of English Language

Small errors and typos need to be worked out.

Author Response

Dear Sir/madam,

Thank you for your valuable comments and inputs to the manuscripts.

Kindly see the attached  our responses for your consideration.

Thanks 

Reviewer 3 Report

Comments and Suggestions for Authors

Dear authors

Thank you for submitting your draft titled "Effectiveness of point of entry health screening measures among travelers in the detection and containment of the international spread of COVID-19: A review of the evidence." I appreciate a lot your effort and  I am grateful to submit my comments and suggest changes to improve the paper.

My main concern is your search strategy could be biases by redundance of your databases selected to explore your main topic:  Scopus database includes the records from the MEDLINE (the same data base of PubMed search engine) and EMBASE databases. I recommend extend your search using other datases like Clarivate Web of Science.

Other main problem are the search terms related to Covid-19... your key words are complex but limited due to the multiple forms found in the database collections. I recommend focusing the search only using terms associated with Covid-19 or SARS-Cov_2 without associating second terms to make the search more extensive and accurate by following the recommendations of:

Lazarus JV, Palayew A, Rasmussen LN, Andersen TH, Nicholson J, Norgaard O. Searching PubMed to Retrieve Publications on the COVID-19 Pandemic: Comparative Analysis of Search Strings. J Med Internet Res. 2020 Nov 26;22(11):e23449. doi: 10.2196/23449. PMID: 33197230; PMCID: PMC7695541.

We invite you to take into consideration our comments and suggestions in the attached  .pdf to materialize  substantial improvements and submit them for consideration for publication.

Author Response

Dear Sir/Madam,

Thank you for your valuable comments and inputs.

Kindly see the attached our responses  for your consideration

We will be happy to receive further guidance and support

Thanks alot.
